# High Filamin a Expression in Adrenocortical Carcinomas Is Associated with a Favourable Tumour Behaviour: A European Multicentric Study

**DOI:** 10.3390/ijms242316573

**Published:** 2023-11-21

**Authors:** Rosa Catalano, Barbara Altieri, Anna Angelousi, Maura Arosio, Francesca Bravi, Letizia Canu, Giorgio A. Croci, Mario Detomas, Emanuela Esposito, Emanuele Ferrante, Stefano Ferrero, Carmina T. Fuss, Gregory Kaltsas, Otilia Kimpel, Laura-Sophie Landwehr, Michaela Luconi, Valentina Morelli, Gabriella Nesi, Emma Nozza, Silviu Sbiera, Andreea L. Serban, Cristina L. Ronchi, Giovanna Mantovani, Erika Peverelli

**Affiliations:** 1Department of Clinical Sciences and Community Health, University of Milan, 20122 Milan, Italy; rosa.catalano@unimi.it (R.C.); maura.arosio@unimi.it (M.A.); francesca.bravi@unimi.it (F.B.); emanuela.esposito1@unimi.it (E.E.); emma.nozza@unimi.it (E.N.); 2Division of Endocrinology and Diabetes, Department of Internal Medicine I, University Hospital, University of Wuerzburg, 97080 Wuerzburg, Germany; altieri_b@ukw.de (B.A.); detomas_m@ukw.de (M.D.); fuss_c@ukw.de (C.T.F.); kimpel_o@ukw.de (O.K.); landwehr_l@ukw.de (L.-S.L.);; 3First Department of Internal Medicine, Laikon General Hospital, Medical School, National and Kapodistrian University of Athens, 11527 Athens, Greece; a.angelousi@gmail.com (A.A.); gregory.kaltsas@gmail.com (G.K.); 451st Department of Propaedeutic Internal Medicine, National University of Athens, 11527 Athens, Greece; 5Endocrinology Unit, Fondazione Istituto di Ricovero e Cura a Carattere Scientifico (IRCCS) Ca’ Granda Ospedale Maggiore Policlinico, 20122 Milan, Italy; emanuele.ferrante@policlinico.mi.it (E.F.); morellivale@yahoo.it (V.M.); andreea.l.serban@gmail.com (A.L.S.); 6Endocrinology Unit, Department of Experimental and Clinical Biomedical Sciences “Mario Serio”, University of Florence, 50139 Florence, Italy; letizia.canu@unifi.it (L.C.); michaela.luconi@unifi.it (M.L.); gabriella.nesi@unifi.it (G.N.); 7Centro di Ricerca e Innovazione sulle Patologie Surrenaliche, AOU Careggi, 50134 Florence, Italy; 8Pathology Unit, Fondazione Istituto di Ricovero e Cura a Carattere Scientifico (IRCCS) Ca’ Granda Ospedale Maggiore Policlinico, 20122 Milan, Italystefano.ferrero@unimi.it (S.F.); 9Ph.D. Program in Experimental Medicine, University of Milan, 20122 Milan, Italy; 10Department of Biomedical, Surgical and Dental Sciences, University of Milan, 20122 Milan, Italy; 11Institute of Metabolism and System Research, University of Birmingham, Birmingham B15 2TT, UK; c.l.ronchi@bham.ac.uk; 12Centre for Endocrinology, Diabetes and Metabolism (CEDAM), Birmingham Health Partners, Birmingham B15 2TT, UK

**Keywords:** filamin A, IGF2, adrenocortical carcinoma, molecular marker, ACC prognostic factor

## Abstract

The insulin-like growth factor 2 (IGF2) promotes cell growth by overactivating the IGF system in an autocrine loop in adrenocortical carcinomas (ACCs). The cytoskeleton protein filamin A (FLNA) acts as a repressor of IGF2 mitogenic signalling in ACC cells. The aims of this study were to test FLNA expression by immunohistochemistry in 119 ACCs and 26 adrenocortical adenomas (ACAs) and to evaluate its relationship with clinicopathological features and outcome in ACCs. We found that 71.4% of ACCs did not express FLNA, whereas FLNA absence was a rare event in ACAs (15.4%, *p* < 0.001 vs. ACCs). In addition, the expression of FLNA was associated with a less aggressive tumour behaviour in ACCs. Indeed, the subgroup of ACCs with high FLNA showed a lower ENSAT stage, Weiss score, and S-GRAS score compared to ACCs with low FLNA expression (*p* < 0.05). Moreover, patients with high FLNA had a longer overall survival than those with low FLNA (*p* < 0.05). In conclusion, our data suggest that FLNA may represent a “protective” factor in ACCs, and the integration of FLNA immunohistochemical expression in ACC tissues along with other clinical and molecular markers could be helpful to improve diagnostic accuracy and prognosis prediction in ACCs.

## 1. Introduction

Adrenocortical tumours affect 3–10% of the population and include a variety of diseases ranging from frequent benign adrenocortical adenomas (ACAs) to rare and generally aggressive adrenocortical carcinomas (ACCs). The differential diagnosis between ACAs and ACCs is based on the Weiss score, which is composed of nine histopathological items (three concerning the tumour structure, three the cytology, and three the invasion) [1]. Although the pathological diagnosis can discriminate benign from malignant tumours in most cases, borderline tumours with a Weiss score of 2 or 3 require further molecular markers for proper classification, in addition to those recently identified such as Type II A topoisomerase (TOP2A), livin, cyclin-dependent kinase 4 (CDK4), and fascin [2,3].

The prognosis of ACCs is generally poor, and the median overall survival (OS) ranges between 3 and 4 years [4,5]. However, the clinical behaviour of ACCs can be very heterogeneous. The main clinical and histopathological prognostic factors are the European Network for the Study of Adrenal Tumors (ENSAT) staging [6] and the Ki67 proliferation index [7], respectively. However, both tumour stage and Ki67 are not always able to discriminate between patients with a good or bad prognosis; therefore, more reliable prognostic factors are required. Recently, a high prognostic performance of the S-GRAS score, which combines the ENSAT stage, Ki67, resection status, age, and presence of symptoms has been demonstrated [8,9,10].

From a pathogenic perspective, several mechanisms have been demonstrated to be involved in adrenal tumorigenesis, including altered regulation of the cell cycle and activation of the insulin-like growth factor (IGF) system [11]. Indeed, IGF2 is overexpressed in 80–90% of ACCs compared to normal adrenals and ACAs [12,13,14,15,16,17,18], with a consequent overactivation of the IGF system that supports the growth of cancer cells in an autocrine loop.

The cytoskeleton actin binding protein filamin A (FLNA) has recently been demonstrated to play an important role in negatively regulating the IGF system in ACC cells [19]. FLNA is a high-molecular-weight multifunctional protein, essential for normal human development, composed of two subunits of 280 kDa each that self-assemble. Each monomer possesses an actin-binding domain at the N-terminus, followed by 24 immunoglobulin-like repeats of about 96 amino acid residues each. FLNA can bind many different partner proteins, including transmembrane proteins and intracellular signalling molecules, acting as a molecular platform that orchestrates a variety of intracellular processes, including cell migration and proliferation [20].

In ACC cell line H295R, FLNA acts as a specific repressor of the IGF2-induced signalling cascade that promotes cell proliferation through extracellular signal-regulated kinases (ERKs) activation. Indeed, genetic silencing of FLNA induced an increase in the insulin-like growth factor 1 receptor (IGF1R) expression and signalling in H295R and ACC primary cultured cells and potentiated cell responsiveness to the IGF1R-insulin receptor (IR) inhibitor linsitinib [19]. Western blot analysis in a small group of adrenocortical tumours indicated that the mean FLNA expression in ACCs was significantly lower than in ACAs [19].

The present study aimed to test the expression of FLNA in a large, multicentre, and well-characterized cohort of ACC and ACA tissues by immunohistochemistry and to test its relationship with the clinicopathological features and outcome, with the final purpose of assessing its relevance as a biomarker to improve the diagnostic accuracy and/or prognostic classification in ACCs.

## 2. Results

### 2.1. Demographic, Clinical, and Histopathological Data

In the ACC group (*n* = 119, Table 1), most of the patients were female (1.77:1), with a median age of 50 (interquartile range (IQR) 38–61) years old. ACC was associated with steroid oversecretion in 64.7% of cases, while 35.3% were non-functional tumours. All patients underwent adrenal surgery as first line treatment and were staged according to the ENSAT system. The median Ki67 index was 15 (IQR 8–23), and the median Weiss score was 6 (IQR 5–7). Full data to calculate the S-GRAS score were available in 55 ACC cases (Table 1).

In total, 85.1% of patients received mitotane postoperatively, and 50% of these were also treated with systemic chemotherapy (Table 1). Data about the clinical outcome were available for 113 ACC, with a median duration of followup of 30 months (IQR 17.5–59.0); six patients were lost to followup and 33.3% of patients died within 60 months after diagnosis.

In the ACA group (*n* = 26, Table 1), we observed a female predominance (1.36:1), with a mean age at diagnosis of 58.0 ± 9.3. All except two tumours presented hormone overproduction. The median duration of followup was 19 months (IQR 4.5–50.5) (21 ACA). In none of the cases was death recorded.

Comparing the ACA and ACC groups, we found a higher age at diagnosis and a prevalence of patients with single hormone secretion in ACA vs. ACC (*p* < 0.05 and *p* < 0.001, respectively).

### 2.2. FLNA Expression in ACC and ACA

Immunohistochemical analysis of FLNA expression in ACC showed the absence of immunoreactivity in 71.4% of the samples (85/119). Representative pictures are shown in Figure 1A.

In tumours with any degree of FLNA expression, positivity was observed mostly in a very heterogeneous fashion. In FLNA-positive tumours, the immunoreactivity score (IRS) was 1 in 31/34 ACCs, while the remaining three ACCs showed a FLNA IRS of 2, 3, and 4, respectively. In total, 18 ACCs showed a percentage of positive cells ≤5%, 13 samples showed a percentage included from 6% to 30%, 2 from 31% to 60%, and 1 from 61% to 100%.

In contrast, 22 out of 26 of ACAs (84.6%) were positive for FLNA (*p* < 0.001 compared to ACC, Figure 1A,B), with an IRS of 1 in 17 ACA, IRS 2 in 4 ACA, and IRS 6 in 1 ACA.

Since FLNA is expected to be localized in various intracellular compartments, we investigated the pattern of FLNA staining (i.e., cytoplasmic and/or membrane positivity). FLNA was localized in the cytoplasm in all positive samples, and in a subgroup of tumours, a plasma membrane localization was also observed (9/34, 26.5% in ACCs and 2/22, 9.1% in ACAs, *p* = n.s.) (Figure 1A).

### 2.3. Association between the FLNA Expression and the Clinicopathological Features of ACCs

No significant differences in clinicopathological features were found between the group of FLNA-positive and -negative ACCs. However, since the group of ACCs considered positive for FLNA expression also included tumours with a very low percentage of immunopositive cells, we divided the ACCs with a percentage of FLN- positive cells less than or equal to 5% (called “low FLNA”, 103/119 patients) from the ACCs with a percentage of positive cells higher than 5% (“high FLNA”, 16/119 patients). We found that the ENSAT stage and Weiss score were significantly lower in the high FLNA group (*p* < 0.05 vs. low FLNA) (Table 2). Indeed, more than 70% of patients belonging to the high FLNA group had an ENSAT stage equal to 1 or 2 and a Weiss score below 6 (*p* < 0.05 vs. low FLNA). In addition, given that the Weiss score is a multiparameter scoring system based on nine histological criteria, we sought to better understand which of them was associated with FLNA expression. We found a reduced mitotic rate in the group with high vs. low FLNA (*p* < 0.05) (Table 3). Moreover, a higher proportion of high FLNA ACCs showed a complete resection status (90% vs. 62.3%, *p* < 0.05). Accordingly, in the high FLNA group, the number of patients with an S-GRAS score higher than 3 was significantly lower than the low FLNA group (*p* < 0.05) (Table 2).

No association was found between the high or low FLNA group and the other clinicopathological features analysed (sex, age, Ki67, hormone secretion, and therapy) (Table 2).

Regarding FLNA intracellular localization, the subgroup of ACCs showing a plasma membrane FLNA immunopositivity did not present any significant difference in the clinicopathological features compared to the subgroup of ACCs with cytoplasmic FLNA.

Lastly, no association between FLNA staining and clinical data in the ACA cohort was found.

### 2.4. Association between FLNA Expression and the Clinical Outcome of ACCs

We then evaluated the potential prognostic role of FLNA expression by univariate survival analysis. The high FLNA group showed a longer OS than the low FLNA group (log rank test, *p* < 0.05) (Figure 2). No differences in progression free survival (PFS) were found between the two groups (log rank test, *p* = 0.225, Appendix A). A multivariate regression analysis revealed a tendency for the significant protective role of FLNA adjusted for the S-GRAS score (*p* = 0.09) (Table 4) and adjusted for the ENSAT stage and Ki67 (*p* = 0.08) (Table 5).

No association was found between the intracellular localization of FLNA and the clinical outcome.

## 3. Discussion

The present study indicates that FLNA expression is lost in the majority of ACCs but not in ACAs, and high FLNA expression in ACCs is associated with a less aggressive tumour behaviour compared to low FLNA.

By analysing FLNA immunoreactivity in 119 ACCs, we found that FLNA was completely absent in 85 samples (71.4%). An opposite situation was found in ACAs, where FLNA was expressed in 22 out of 26 samples. These observations are in line with previously published data analysing FLNA protein expression by Western blot in a small group of ACC patients [19]. In other human tumour types, FLNA has been found overexpressed or downregulated. In fact, an overexpressed FLNA was found in multiple types of cancer, including prostate [21], breast [22,23], lung cancer [24], squamous cell carcinoma [25], hepatic cholangiocarcinoma [26], parathyroid carcinomas [27], cervical cancer [28] and pulmonary neuroendocrine tumours [29]. In contrast, FLNA was found to be downregulated in a small number of cancer types, including colorectal cancer [30], bladder carcinoma [31], nasopharyngeal carcinoma [32], and gastric carcinoma [33]. Our data show that ACCs belong to the latter subgroup of cancers.

The observation that FLNA absence is typical of ACCs, while it is a rare event in ACAs, suggests the possible use of FLNA expression as an immunohistochemical marker to discriminate ACCs from ACAs in those situations where histologic differentiation could be challenging [34,35,36]. Moreover, although the potential malignant evolution of ACAs into ACCs is still under debate [37,38], we can hypothesize that the absence of FLNA in some ACAs may represent a predisposing factor to a following premalignant state. In support of this hypothesis, FLNA silencing in primary ACA cell cultures, derived from human surgically removed ACAs, was found to increase both ERK phosphorylation and cell proliferation [19].

In ACCs, the sole presence of FLNA was not associated with any clinicopathological features. However, we observed that the group of ACCs considered positive for FLNA expression included tumours with a very low percentage of immunopositive cells. We hypothesized that the effect of FLNA in these tumours could be negligible. We thus divided ACCs based on the percentage of FLNA-positive cells less than or equal to 5% (called “low FLNA”) from ACCs with a percentage of positive cells higher than 5% (“high FLNA”). We found that the small group of high FLNA ACC displayed less aggressive features, with a lower ENSAT stage, Weiss score, and S-GRAS score, and longer survival than ACCs expressing low FLNA levels. Moreover, we demonstrated that FLNA expression was associated also with the individual Weiss parameter mitotic rate, with a lower number of mitoses in the high FLNA group. This finding agrees with the previously demonstrated role of FLNA in decreasing IGF2 mitogenic effects in ACC cells through a reduction in ERK phosphorylation [19]. To note, although not significant, most patients (75%) with high FLNA did not present necrosis compared to 33% of the low FLNA group, confirming again the potential protective role of FLNA on ACC aggressiveness. Indeed, it has been recently demonstrated that necrosis is the best histological survival predictor in ACCs among the Weiss parameters [39].

Interestingly, we found that high FLNA ACC patients showed a longer OS compared to low FLNA ACCs. However, in the multivariate survival model performed with FLNA adjusted for S-GRAS, we observed only a tendency toward the protective role of FLNA, the lack of significance being probably due to the small sample size. Moreover, it is important to note that after adjustment for the ENSAT tumour stage and Ki67, even if not significant, the FLNA protective hazard ratio did not change (HR = 0.2).

In the literature, the role of FLNA expression in the OS of patients varies according to the type of tumours. High levels of FLNA were associated with longer OS in colorectal cancer [30], while in gastric cancer [33] and melanoma [40] FLNA expression is associated with poor OS, corroborating the idea that the effects of FLNA are cancer-type specific.

We can hypothesize that in the absence of FLNA, IGF2 mitotic effects are potentiated, leading to a final worse outcome in ACCs. Indeed, we previously demonstrated that FLNA is able to bind IGF1R, and FLNA silencing induced a reduction in the IGF1R expression and a simultaneous decrease in ERK phosphorylation, cyclin E1 expression, and cell proliferation in human ACC cell line H295R and in primary cultured ACC cells [19], supporting a role for FLNA as a suppressor of IGF2 signalling.

Our previous and current findings suggest that FLNA may represent an alternative drug target along the IGF2/IGF1R pathway, possibly overcoming the failure of IGF1R inhibitors in clinical trials. Therapeutic approaches aimed to restore FLNA expression could represent a novel strategy in the treatment of ACCs.

Admittedly, the evaluation of FLNA expression detected by immunohistochemistry does not take into account posttranslational modifications that are crucial in determining FLNA activity. Since one of the main mechanisms that control FLNA functions and proteolysis is phosphorylation on Ser2152 [41,42,43], further studies evaluating FLNA phosphorylation levels in ACCs are needed to further understand the contribute of FLNA to the reduction in cancer progression.

In conclusion, the present study indicates that FLNA is absent in the majority of ACCs but not in ACAs, and high FLNA expression is associated with a less aggressive tumour behaviour compared to low FLNA in ACCs. Overall these data suggest that FLNA may be a cancer suppressor factor in adrenocortical tumours. The integration of FLNA expression assessment with other clinical and molecular markers will be helpful to improve both the diagnostic accuracy and prognosis prediction in ACCs.

## 4. Materials and Methods

### 4.1. Patients and Data Collection

This is a retrospective European multicentre study conducted on behalf of the ENSAT (www.ensat.org). In total, 119 patients, who underwent adrenalectomy because of ACCs, were included in the study, i.e., 74 patients from the University Hospital of Wuerzburg (Germany), 29 from the University of Florence (Italy), and 16 from Laiko Hospital Athens (Greece). Moreover, 26 tissue sections derived from patients with histological diagnosis of ACA that had surgery in Foundation IRCCS Ca’ Granda Ospedale Maggiore Policlinico were analysed.

ACC patients’ age, sex, date of diagnosis, ENSAT tumour stage at diagnosis, Ki67 proliferation index, Weiss score, tumour resection (R) status, tumour-related hormone excess at diagnosis, symptoms, date of last followup, and status of patient (alive or date of death) were recorded. The S-GRAS score for the ACC samples was calculated as previously published [9], resulting in S-GRAS scores ranging from 0 to 9. Details about medical treatments, including administration of adjuvant mitotane treatment and/or chemotherapy were also collected from ACC patients’ medical records. Clinical outcome (including overall and progression-free survival) was evaluated by cross-sectional imaging and clinical judgment. Overall survival (OS) was defined as the time from primary tumour resection or diagnosis to death or last followup. Progression-free survival (PFS) was evaluated only in a subgroup of 74 patients and was defined as the time from diagnosis to first radiological evidence of disease progression. Specifically, radiological evidence of progression or relapse was detected during periodical radiological surveillance performed every three months by thorax–abdomen–pelvis computed tomography scan with contrast (TAP CT scan). The last followup was November 2020.

Moreover, a database for ACA patients with the following data were created: sex, age, date of diagnosis, histopathological characteristics, tumour-related hormone excess, date of last followup, and status of patient.

The study was approved by the local ethics committees, and each patient gave written informed consent to the use of their tumour sample and clinical information.

### 4.2. Immunohistochemistry

Slides of 4 µm thickness were obtained from representative formalin-fixed paraffin-embedded (FFPE) blocks of adrenocortical tumours (119 ACC and 26 ACA). Particularly, only primary tumours were investigated.

Immunohistochemical staining with anti-FLNA antibody (polyclonal, Catalog ID LS-B4865, LifeSpan Bioscience, Seattle, WA, USA; 0.42 mg/mL concentration, 1:150 dilution) was performed using an automated stainer (BenchMark ULTRA, Ventana-Roche diagnostics, Oro Balley, AZ, USA), after EDTA buffer retrieval and via diaminobenzidine revelation (Ultraview Universal DAB Detection Kit, Ventana-Roche diagnostics). The internal positive control for proper staining was represented by stromal cells. The following parameters were assessed by 2 independent investigators: percentage of positive tumoral cells; pattern of staining (i.e., cytoplasmic and/or membrane positivity); intensity of staining. Discordant results were reviewed and discussed among the 2 investigators until a consensus was reached. Immunoreactivities were graded manually using an optical microscope (Eclipse Ci-E, Nikon Instruments Inc., Tokyo, Japan) according to an immunoreactivity score (IRS) that takes into account both the percentage of positive cells (0–30% = 1; 31–60% = 2; 61–100% = 3) and the staining intensity (0 = absence of immunoreactivity, 1 = weak, 2 = medium intensity, and 3 = strong reactivity), with a final score being obtained by multiplying both parameters together (range 0–9). Representative scans were captured via an Aperio AT-2 scanner (Leica Biosystems, Wetzlar, Germany). Tissues with FLNA staining >5% of positive cells were considered “high FLNA”, whereas those with FLNA staining ≤5% of cells were considered “low FLNA”.

### 4.3. Statistical Analyses

Continuous variables were reported as the mean ± standard deviation (SD) or median and IQR, according to their distribution, and were compared using parametric or non-parametric tests, respectively. Discrete variables were described as numbers and percentages and compared through the chi-square test or Fisher’s exact test, when appropriate. The Kaplan–Meyer method was used to describe the OS and PFS according to FLNA expression and different patient characteristics. The log-rank test was used to test the difference in survival across groups. The prognostic role of FLNA was also assessed in multivariate analysis according to the Cox regression model. We evaluated the potential prognostic role of FLNA adjusted for the S-GRAS score since it was recently demonstrated to have a superior prognostic performance respect to tumour stage and Ki67 in ACCs [9]. Moreover, due to the limited number of patients with all data required to calculate the S-GRAS score, we also performed a multivariate survival model including as covariates ENSAT stage and Ki67.

Statistical analyses were carried out using the SAS software, version 9.4 (SAS Institute, Inc., Cary, NC, USA), and IBM SPSS statistics, version 28.0.1.1 (SPSS Inc., Chicago, IL, USA).

## Figures and Tables

**Figure 1 ijms-24-16573-f001:**
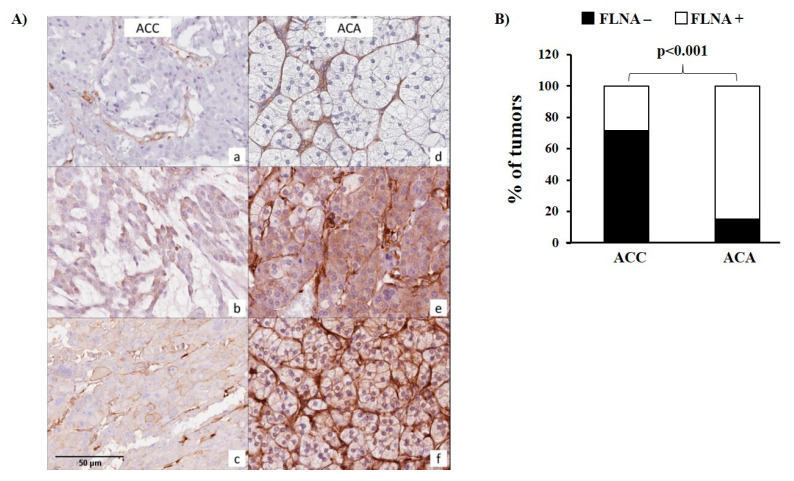
Filamin A (FLNA) protein staining and cellular localization in ACC and ACA. (**A**) Panels on the left depict, respectively, (**a**) the negative pattern, with positivity on the stromal component as internal control; (**b**) the mostly weak cytoplasmic staining; and (**c**) the membranous enhancement in representative cases of ACC. The right side documents (**d**) negative staining; (**e**) cytoplasmic positivity; and (**f**) membrane staining in selected ACA cases. (**B**) The bar graph shows the comparison of the percentage of tumours expressing or not FLNA between ACC and ACA. Statistical analysis was performed using the chi-square test, *p* < 0.001.

**Figure 2 ijms-24-16573-f002:**
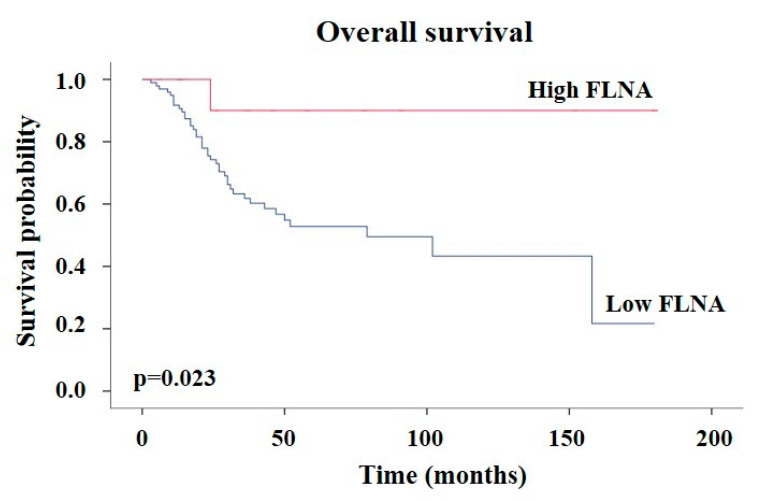
ACCs expressing high FLNA levels have a better survival than the low FLNA group. Kaplan–Meier analysis of the overall survival of ACC patients expressing low (*n* = 98) and high (*n* = 14) FLNA levels.

**Table 1 ijms-24-16573-t001:** Demographic, clinical, and histopathological characteristics of patients with adrenocortical carcinoma (ACC) and adenoma (ACA).

	ACC (*n* = 119)	ACA (*n* = 26)	
	Number (%)	Number (%)	*p* Value
**Sex**			
Female/Male (%)	76/43 (63.9/36.1)	15/11 (57.7/42.3)	0.55 *
**Age**			
≥50 years	60 (50.4)	17 (81.0)	***0.02*** ^−^
**Hormonal overproduction**			
Inactive	36 (35.3)	2 (8.0)	***0.0001*** ^−^
Single hormone	38 (37.3)	19 (76.0)	
More than one hormone	28 (27.4)	4 (16.0)	
**ENSAT tumour stage**			
1	17 (14.4)	/	
2	44 (37.3)	/	
3	40 (33.9)	/	
4	17 (14.4)	/	
**Ki67**			
0–9%	30 (27.0)	/	
10–19%	38 (34.2)	/	
≥20%	43 (38.7)	/	
**Weiss score**			
<6	40 (44.0)	/	
≥6	51 (56.0)	/	
**Resection status**			
R0	47 (66.2)	/	
R1	7 (9.9)	/	
R2	9 (12.7)	/	
RX	8 (11.3)	/	
**S-GRAS score**			
0–1	8 (14.5)	/	
2–3	22 (40.0)	/	
4–5	14 (25.5)	/	
6–9	11 (20.0)	/	
**Post-surgical therapy**			
No medical treatment	17 (14.9)	/	
Mitotane	40 (35.1)	/	
Mitotane + chemotherapy	57 (50.0)	/	

* *p*-value obtained from chi-square test; ^−^ *p*-value obtained from Fisher’ exact test.

**Table 2 ijms-24-16573-t002:** Clinicopathological features in the low FLNA and high FLNA groups of ACCs.

	Low FLNA (*n* = 103)	High FLNA (*n* = 16)	*p* Value
**Female (%)**	62.1	75.0	0.32 *
**Age**	49.7 ± 14.8	50.9 ± 14.9	0.76 ^+^
**≥50 years (%)**	49.5	56.3	0.62 *
**ENSAT stage (1–2) (%)**	48.0	75.0	***0.04*** *
**ENSAT stage (3–4) (%)**	52.0	25.0	
**Ki67**	18.8 ± 16.4	13.0 ± 10.6	0.20 ^+^
**Ki67 < 10 (%)**	24.7	42.9	0.15 *
**Ki67 ≥ 10 (%)**	75.3	57.1	
**Weiss < 6 (%)**	39.0	71.4	***0.02*** *
**Weiss ≥ 6 (%)**	61.0	28.6	
**Resection status (%)**			
**R0**	62.3	90.0	***0.02*** ^−^
**R1**	11.5	0.0	
**R2**	14.8	0.0	
**RX**	11.5	10.0	
**S-GRAS 0–1 (%)**	10.6	37.5	***0.02*** ^−^
**S-GRAS 2–3 (%)**	40.4	37.5	
**S-GRAS > 3 (%)**	48.9	25.0	
**Secreting tumours (%)**	64.4	66.7	0.86 *
**Non-secreting tumours (%)**	35.6	33.3	
**Mitotane chemotherapy (%)**	49.0	56.3	0.24 *

* *p*-value obtained from chi-square test; ^+^ *p*-value obtained from *t*-test; ^−^ *p*-value obtained from Fisher’ exact test.

**Table 3 ijms-24-16573-t003:** The individual histological Weiss parameters in the low and high FLNA groups of ACCs.

Histological Weiss Criteria	Low FLNA (%)	High FLNA (%)	*p* Value
**Nuclear grade**			
High	20.8	25.0	1.00
Low	79.2	75.0	
**Mitoses**			
≤5 per 50 high-power fields	16.7	75.0	** *0.04* **
>5 per 50 high-power fields	83.3	25.0	
**Atypical mitoses**			
No	70.8	75.0	1.00
Yes	29.2	25.0	
**Clear cells**			
>25%	8.3	0.0	1.00
≤25%	91.7	100.0	
**Diffuse architecture**			
≤33% surface	4.2	0.0	1.00
>33% surface	95.8	100.0	
**Confluent necrosis**			
No	33.3	75.0	0.27
Yes	66.7	25.0	
**Venous invasion**			
No	62.5	75.0	1.00
Yes	37.5	25.0	
**Sinusoidal invasion**			
No	41.7	75.0	0.31
Yes	58.3	25.0	
**Capsular infiltration**			
No	29.2	75.0	0.12
Yes	70.8	25.0	

*p*-values obtained from Fisher’s exact test.

**Table 4 ijms-24-16573-t004:** Univariate and multivariate Cox regression analysis for risk of death in high vs. low FLNA ACCs adjusted for S-GRAS.

Variable	Univariate				Multivariate			
	*n*	*HR*	*95%CI*	*p*	*n*	*HR*	*95%CI*	*p*
**FLNA expression**	112	0.2	[0.2–1.0]	0.05	55	0.2	[0.2–1.3]	0.09
Low	98				47			
High	14				8			

HR: hazard ratio; CI: confidence interval; *n*: number of evaluated patients.

**Table 5 ijms-24-16573-t005:** Univariate and multivariate Cox regression analysis for risk of death in the ACC cohort.

Variables	Univariate				Multivariate			
	*n*	*HR*	*95%CI*	*p*	*n*	*HR*	*95%CI*	*p*
**FLNA expression**	112	0.2	[0.2–1.0]	0.05	105	0.2	[0.2–1.2]	0.08
Low	98				92			
High	14				13			
**ENSAT stage**	111	3.9	[2.0–7.7]	** *<0.001* **	105	4.6	[2.2–9.6]	** *<0.001* **
1–3	96				93			
4	15				12			
**Ki67**	105	4.5	[1.6–12.8]	** *0.005* **	105	3.0	[1.0–8.7]	** *0.04* **
<10	30				30			
≥10	75				75			

HR: hazard ratio; CI: confidence interval; *n*: number of evaluated patients.

## Data Availability

The data presented in this study are available on request from the corresponding author. The data are not publicly available due to privacy (patient data).

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
