# Peer review of "High Filamin a Expression in Adrenocortical Carcinomas Is Associated with a Favourable Tumour Behaviour: A European Multicentric Study"

_ijms, 2023, doi:10.3390/ijms242316573_

Round 1
Reviewer 1 Report
Comments and Suggestions for Authors
In the current manuscript entitled ‘High filamin A expression in adrenocortical carcinomas is 2 associated with a favourable tumor behaviour: a European multicentric study’, Catalano and colleagues are assessing the expression of filamin A by immunohistochemistry in ACC and ACA samples and investigate the correlation of its expression with clinicopathological features and outcomes. The introduction is clear, concise and well written. It gives the information necessary to understand the main objective of the manuscript. I only have few comments for the results section detailed below.
1. Since the main results of the manuscript rely on the expression of filamin A, it would be important to provide some details about the validation of the antibody. The use of a blocking peptide on a couple of representative samples would help validate the specificity of the antibody by immunohistochemistry.
2. The authors suggest that FLNA staining could be integrated as a molecular marker for diagnostic, therefore it is crucial to provide more details about the protocol used, such as the catalog number for the antibody, perhaps the concentration used in ug/ml, in addition to the dilution. Neither the antigen retrieval, the type of secondary antibody, or the chromogenic substrate used are mentioned in the methods section.
3. Figure 1, scale bars would be more appropriate rather than the magnification in the legends.
4. The authors describe the expression of FLNA either at the membrane or in the cytoplasm. Is this protein expected to be localized in various location, what is the molecular basis ?
5. The authors demonstrated that filamin A is not expressed in 71% of ACC samples and mentioned a ‘loss’ of expression but it is unclear in the manuscript if the expression has been described in the normal human adrenal, and if so which cell population/zone is marked. If these data exist in the literature, references could be simply cited.
Author Response
REVIEWER 1
We thank the reviewer for the careful reading of our manuscript and for her/his comments and suggestions. The issues raised by the reviewer have been carefully addressed.
Since the main results of the manuscript rely on the expression of filamin A, it would be important to provide some details about the validation of the antibody. The use of a blocking peptide on a couple of representative samples would help validate the specificity of the antibody by immunohistochemistry.
We thank the reviewer for the comment. The immunohistochemical assessment was performed according to the specifications of the datasheet of the selected FLNA clone and the specificity of the reaction was assessed by identification of the proper internal control (i.e. stromal component).
The authors suggest that FLNA staining could be integrated as a molecular marker for diagnostic, therefore it is crucial to provide more details about the protocol used, such as the catalog number for the antibody, perhaps the concentration used in ug/ml, in addition to the dilution. Neither the antigen retrieval, the type of secondary antibody, or the chromogenic substrate used are mentioned in the methods section.
We thank the reviewer for the comment. The methods section has been edited by providing the ID of the selected clone and its concentration and the retrieval specification. As to the secondary antibody and chromogenic substrate, they correspond to a commercial kit by Roche Ventana, which has been now specified in the revised version of the manuscript as follows: “Immunohistochemical staining with anti-FLNA antibody (polyclonal, Catalog ID LS-B4865, LifeSpan Bioscience, Seattle, WA, USA; 0.42 mg / ml concentration, 1:150 dilution) was performed using an automated stainer (BenchMark ULTRA, Ventana-Roche diagnostics), after EDTA buffer retrieval and via diaminobenzidine revelation (Ultraview Universal DAB Detection Kit, Ventana-Roche diagnostics).”
(Materials and Methods section, Lines 335-339).
Figure 1, scale bars would be more appropriate rather than the magnification in the legends.
We agree with the reviewer and we added the scale bar accordingly (Revised Figure 1A). Magnification was removed from the legend (Revised Legend Fig.1A lines 158-161).
The authors describe the expression of FLNA either at the membrane or in the cytoplasm. Is this protein expected to be localized in various location, what is the molecular basis?
FLNA is expected to be localized in various intracellular compartments. In particular, it is localised in the cytoplasm where it executes the majority of its functions dimerizing and orthogonally crosslinking actin through its N-terminal actin-binding domain. Moreover, it can be found at the cell membrane where bringing together the transmembrane receptors, the submembrane actin network and the intracellular signalling components, facilitates the activation of local cellular processes (Stossel TP et al. Filamins as integrators of cell mechanics and signalling. Nat Rev Mol Cell Biol. 2001. 2(2):138-45. doi: 10.1038/35052082).
We have added a sentence in the revised Results section 2.2 “Since FLNA is expected to be localized in various intracellular compartments, we investigated pattern of FLNA staining (i.e. cytoplasmic and / or membrane positivity)” (Lines 150-152).
The authors demonstrated that filamin A is not expressed in 71% of ACC samples and mentioned a ‘loss’ of expression but it is unclear in the manuscript if the expression has been described in the normal human adrenal, and if so which cell population/zone is marked. If these data exist in the literature, references could be simply cited.
We thank the reviewer for the comment. No information about FLNA expression in the normal human adrenal are present in literature, therefore we substituted the word “loss” with “absence” (Revised Abstract line 52; Revised Discussion line 243).
Reviewer 2 Report
Comments and Suggestions for Authors
In this study, the authors investigated the expression of Filamin A, a cytoskeletal actin-binding protein, in adrenocortical carcinomas and its potential utility as a biomarker in ACC. The authors adeptly integrated relevant references and background literature into their study. Here are my specific observations:
• In the abstract, it would be beneficial to provide a definition for ENSAT.
• Since the sample predominantly consisted of females, it is advisable for the authors to address the potential impact of gender differences on the study's outcomes.
• Filamin A expression is influenced by a variety of proteins and serves a dual role. To substantiate that Filamin A contributes to the reduction of cancer progression, the authors should supply additional data or explore the possibility of quantifying other biomarkers, such as filamin A phosphorylation or proteins that lead to filamin A degradation.
• In a prior publication, the authors showcased an elevation in IGF1R expression levels upon Filamin A was silenced. Therefore, they should furnish data or engage in a more in-depth discussion regarding how the decrease in Filamin A expression could affect IGF1R signaling in adrenocortical carcinomas (ACC).
Round 2
Reviewer 2 Report
Comments and Suggestions for Authors
The comments are addressed.